# CROSS-NODE FEDERATED GRAPH NEURAL NETWORK FOR SPATIO-TEMPORAL DATA MODELING

## ABSTRACT

Vast amount of data generated from networks of sensors, wearables, and the Internet of Things (IoT) devices underscores the need for advanced modeling techniques that leverage the spatio-temporal structure of decentralized data due to the need for edge computation and licensing (data access) issues. While federated learning (FL) has emerged as a framework for model training without requiring direct data sharing and exchange, effectively modeling the complex spatio-temporal dependencies to improve forecasting capabilities still remains an open problem. On the other hand, state-of-the-art spatio-temporal forecasting models assume unfettered access to the data, neglecting constraints on data sharing. To bridge this gap, we propose a federated spatio-temporal model – Cross-Node Federated Graph Neural Network (CNFGNN) – which explicitly encodes the underlying graph structure using graph neural network (GNN)-based architecture under the constraint of cross-node federated learning, which requires that data in a network of nodes is generated locally on each node and remains decentralized. CNFGNN operates by disentangling the temporal dynamics modeling on devices and spatial dynamics on the server, utilizing alternating optimization to reduce the communication cost, facilitating computations on the edge devices. Experiments on the traffic flow forecasting task show that CNFGNN achieves the best forecasting performance in both transductive and inductive learning settings with no extra computation cost on edge devices, while incurring modest communication cost.

## 1 INTRODUCTION

Modeling the dynamics of spatio-temporal data generated from networks of edge devices or nodes (e.g. sensors, wearable devices and the Internet of Things (IoT) devices) is critical for various applications including traffic flow prediction (Li et al., 2018; Yu et al., 2018), forecasting (Seo et al., 2019; Azencot et al., 2020), and user activity detection (Yan et al., 2018; Liu et al., 2020). While existing works on spatio-temporal dynamics modeling (Battaglia et al., 2016; Kipf et al., 2018; Battaglia et al., 2018) assume that the model is trained with centralized data gathered from all devices, the volume of data generated at these edge devices precludes the use of such centralized data processing, and calls for decentralized processing where computations on the edge can lead to significant gains in improving the latency. In addition, in case of spatio-temporal forecasting, the edge devices need to leverage the complex inter-dependencies to improve the prediction performance. Moreover, with increasing concerns about data privacy and its access restrictions due to existing licensing agreements, it is critical for spatio-temporal modeling to utilize decentralized data, yet leveraging the underlying relationships for improved performance.

Although recent works in federated learning (FL) (Kairouz et al., 2019) provides a solution for training a model with decentralized data on multiple devices, these works either do not consider the inherent spatio-temporal dependencies (McMahan et al., 2017; Li et al., 2020b; Karimireddy et al., 2020) or only model it implicitly by imposing the graph structure in the regularization on model weights (Smith et al., 2017), the latter of which suffers from the limitation of regularization based methods due to the assumption that graphs only encode similarity of nodes (Kipf & Welling, 2017), and cannot operate in settings where only a fraction of devices are observed during training (*inductive learning setting*). As a result, there is a need for an architecture for spatio-temporal data modeling which enables reliable computation on the edge, while maintaining the data decentralized.

To this end, leveraging recent works on federated learning (Kairouz et al., 2019), we introduce the *cross-node* federated learning requirement to ensure that data generated locally at a node remains decentralized. Specifically, our architecture – Cross-Node Federated Graph Neural Network (CN-FGNN), aims to effectively model the complex spatio-temporal dependencies under the cross-node federated learning constraint. For this, CNFGNN decomposes the modeling of temporal and spatial dependencies using an encoder-decoder model on each device to extract the temporal features with local data, and a Graph Neural Network (GNN) based model on the server to capture spatial dependencies among devices.

As compared to existing federated learning techniques that rely on regularization to incorporate spatial relationships, CNFGNN leverages an explicit graph structure using a graph neural network-based (GNNs) architecture, which leads to performance gains. However, the federated learning (data sharing) constraint means that the GNN cannot be trained in a centralized manner, since each node can only access the data stored on itself. To address this, CNFGNN employs Split Learning (Singh et al., 2019) to train the spatial and temporal modules. Further, to alleviate the associated high communication cost incurred by Split Learning, we propose an alternating optimization-based training procedure of these modules, which incurs only half the communication overhead as compared to a comparable Split Learning architecture. Here, we also use Federated Averaging (FedAvg) (McMahan et al., 2017) to train a shared temporal feature extractor for all nodes, which leads to improved empirical performance.

Our main contributions are as follows :

1. We propose Cross-Node Federated Graph Neural Network (CNFGNN), a GNN-based federated learning architecture that captures complex spatio-temporal relationships among multiple nodes while ensuring that the data generated locally remains decentralized at no extra computation cost at the edge devices.

2. Our modeling and training procedure enables GNN-based architectures to be used in federated learning settings. We achieve this by disentangling the modeling of local temporal dynamics on edge devices and spatial dynamics on the central server, and leverage an alternating optimization-based procedure for updating the spatial and temporal modules using Split Learning and Federated Averaging to enable effective GNN-based federated learning.

3. We demonstrate that CNFGNN achieves the best prediction performance (both in transductive and inductive settings) at no extra computation cost on edge devices with modest communication cost, as compared to the related techniques on a traffic flow prediction task.

## 2 RELATED WORK

Our method derives elements from graph neural networks, federated learning and privacy-preserving graph learning, we now discuss related works in these areas in relation to our work.

**Graph Neural Networks (GNNs).** GNNs have shown their superior performance on various learning tasks with graph-structured data, including graph embedding (Hamilton et al., 2017), node classification (Kipf & Welling, 2017), spatio-temporal data modeling (Yan et al., 2018; Li et al., 2018; Yu et al., 2018) and multi-agent trajectory prediction (Battaglia et al., 2016; Kipf et al., 2018; Li et al., 2020a). Recent GNN models (Hamilton et al., 2017; Ying et al., 2018; You et al., 2019; Huang et al., 2018) also have sampling strategies and are able to scale on large graphs. While GNNs enjoy the benefit from strong inductive bias (Battaglia et al., 2018; Xu et al., 2019), most works require centralized data during the training and the inference processes.

**Federated Learning (FL).** Federated learning is a machine learning setting where multiple clients train a model in collaboration with decentralized training data (Kairouz et al., 2019). It requires that the raw data of each client is stored locally without any exchange or transfer. However, the decentralized training data comes at the cost of less utilization due to the heterogeneous distributions of data on clients and the lack of information exchange among clients. Various optimization algorithms have been developed for federated learning on non-IID and unbalanced data (McMahan et al., 2017; Li et al., 2020b; Karimireddy et al., 2020). Smith et al. (2017) propose a multi-task learning framework that captures relationships amongst data. While the above works mitigate the caveat of missing

neighbors' information to some extent, they are not as effective as GNN models and still suffer from the absence of feature exchange and aggregation.

**Alternating Optimization.** Alternating optimization is a popular choice in non-convex optimization (Agarwal et al., 2014; Arora et al., 2014; 2015; Jain & Kar, 2017). In the context of Federated Learning, Liang et al. (2020) uses alternating optimization for learning a simple global model and reduces the number of communicated parameters, and He et al. (2020) uses alternating optimization for knowledge distillation from server models to edge models. In our work, we utilize alternating optimization to effectively train on-device modules and the server module jointly, which captures temporal and spatial relationships respectively.

**Privacy-Preserving Graph Learning.** Suzumura et al. (2019) and Mei et al. (2019) use statistics of graph structures instead of node information exchange and aggregation to avoid the leakage of node information. Recent works have also incorporated graph learning models with privacy-preserving techniques such as Differential Privacy (DP), Secure Multi-Party Computation (MPC) and Homomorphic Encryption (HE). Zhou et al. (2020) utilize MPC and HE when learning a GNN model for node classification with vertically split data to preserve silo-level privacy instead of node-level privacy. Sajadmanesh & Gatica-Perez (2020) preprocesses the input raw data with DP before feeding it into a GNN model. Composing privacy-preserving techniques for graph learning can help build federated learning systems following the privacy-in-depth principle, wherein the privacy properties degrade as gracefully as possible if one technique fails (Kairouz et al., 2019).

## 3 CROSS-NODE FEDERATED GRAPH NEURAL NETWORK

### 3.1 PROBLEM FORMULATION

Given a dataset with a graph $\mathcal{G} = (\mathcal{V}, \mathcal{E})$, a feature tensor $\mathbf{X} \in \mathbb{R}^{|\mathcal{V}| \times \cdots}$ and a label tensor $\mathbf{Y} \in \mathbb{R}^{|\mathcal{V}| \times \cdots}$, we consider learning a model under the cross-node federated learning constraint: node feature $\boldsymbol{x}_i = \mathbf{X}_{i,\dots}$, node label $\boldsymbol{y}_i = \mathbf{Y}_{i,\dots}$, and model output $\hat{\boldsymbol{y}}_i$ are only visible to the node $i$.

One typical task that requires the cross-node federated learning constraint is the prediction of spatio-temporal data generated by a network of sensors. In such a scenario, $\mathcal{V}$ is the set of sensors and $\mathcal{E}$ describes relations among sensors (e.g. $e_{ij} \in \mathcal{E}$ if and only if the distance between $v_i$ and $v_j$ is below some threshold). The feature tensor $\boldsymbol{x}_i \in \mathbb{R}^{m \times D}$ represents the $i$-th sensor's records in the $D$-dim space during the past $m$ time steps, and the label $\boldsymbol{y}_i \in \mathbb{R}^{n \times D}$ represents the $i$-th sensor's records in the future $n$ time steps. Since records collected on different sensors owned by different users/organizations may not be allowed to be shared due to the need for edge computation or licensing issues on data access, it is necessary to design an algorithm modeling the spatio-temporal relation without any direct exchange of node-level data.

### 3.2 PROPOSED METHOD

We now introduce our proposed Cross-Node Federated Graph Neural Network (CNFGNN) model. Here, we begin by disentangling the modeling of node-level temporal dynamics and server-level spatial dynamics as follows: (i) (Figure 1c) on each node, an encoder-decoder model extracts temporal features from data on the node and makes predictions; (ii) (Figure 1b) on the central server, a Graph Network (GN) (Battaglia et al., 2018) propagates extracted node temporal features and outputs node embeddings, which incorporate the relationship information amongst nodes. (i) has access to the not shareable node data and is executed on each node locally. (ii) only involves the upload and download of smashed features and gradients instead of the raw data on nodes. This decomposition enables the exchange and aggregation of node information under the cross-node federated learning constraint.

### 3.2.1 MODELING OF NODE-LEVEL TEMPORAL DYNAMICS

We modify the Gated Recurrent Unit (GRU) based encoder-decoder architecture in (Cho et al., 2014) for the modeling of node-level temporal dynamics on each node. Given an input sequence $\boldsymbol{x}_i \in \mathbb{R}^{m \times D}$ on the $i$-th node, an encoder sequentially reads the whole sequence and outputs the

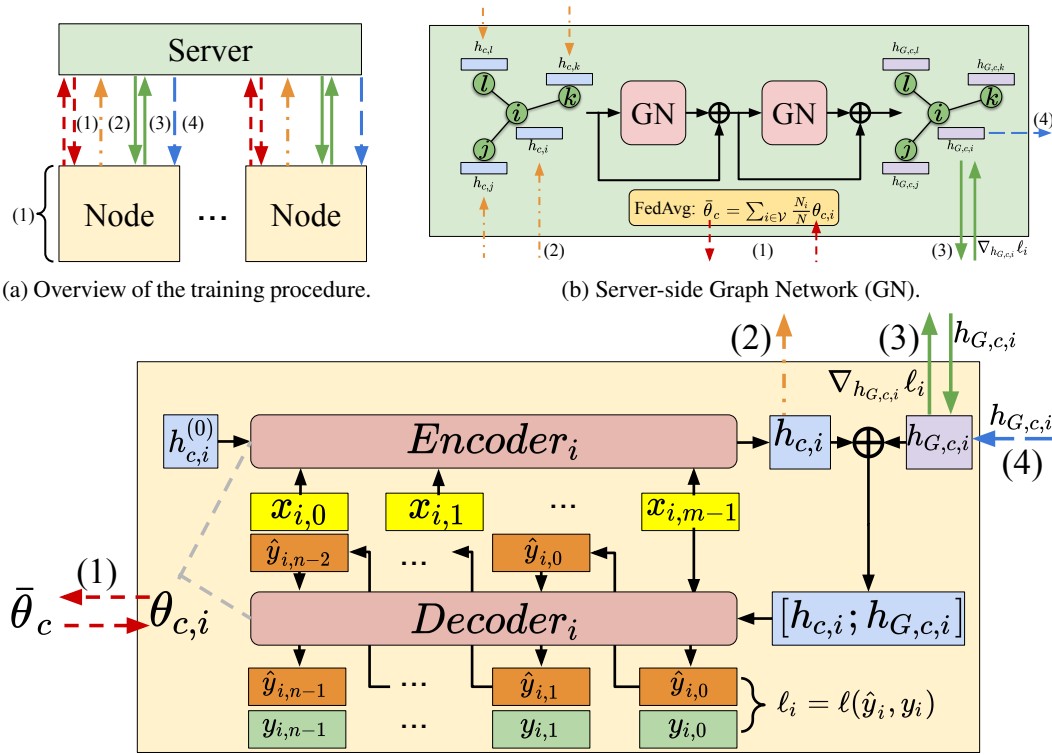

(a) Overview of the training procedure.

(b) Server-side Graph Network (GN).

(c) Encoder-decoder on the $i$-th node.

Figure 1: Cross-Node Federated Graph Neural Network. (a) In each round of training, we alternately train models on nodes and the model on the server. More specifically, we sequentially execute: (1) Federated learning of on-node models. (2) Temporal encoding update. (3) Split Learning of GN. (4) On-node graph embedding update. (b) Detailed view of the server-side GN model for modeling spatial dependencies in data. (c) Detailed view of the encoder-decoder model on the $i$-th node.

hidden state $\boldsymbol{h}_{c,i}$ as the summary of the input sequence according to Equation 1.

$$\boldsymbol{h}_{c,i} = Encoder_i(\boldsymbol{x}_i, \boldsymbol{h}_{c,i}^{(0)}), \tag{1}$$

where $\boldsymbol{h}_{c,i}^{(0)}$ is a zero-valued initial hidden state vector.

To incorporate the spatial dynamics into the prediction model of each node, we concatenate $\boldsymbol{h}_{c,i}$ with the node embedding $\boldsymbol{h}_{G,c,i}$ generated from the procedure described in 3.2.2, which contains spatial information, as the initial state vector of the decoder. The decoder generates the prediction $\hat{\boldsymbol{y}}_i$ in an auto-regressive way starting from the last frame of the input sequence $x_{i,m}$ with the concatenated hidden state vector.

$$\hat{\boldsymbol{y}}_i = Decoder_i(x_{i,m}, [\boldsymbol{h}_{c,i}; \boldsymbol{h}_{G,c,i}]). \tag{2}$$

We choose the mean squared error (MSE) between the prediction and the ground truth values as the loss function, which is evaluated on each node locally.

### 3.2.2 MODELING OF SPATIAL DYNAMICS

To capture the complex spatial dynamics, we adopt Graph Networks (GNs) proposed in (Battaglia et al., 2018) to generate node embeddings containing the relational information of all nodes. The central server collects the hidden state from all nodes $\{\boldsymbol{h}_{c,i} \mid i \in \mathcal{V}\}$ as the input to the GN. Each layer of GN updates the input features as follows:

$$
\begin{array}{ll}
\mathbf{e}'_k = \phi^e\left(\mathbf{e}_k, \mathbf{v}_{r_k}, \mathbf{v}_{s_k}, \mathbf{u}\right) & \overline{\mathbf{e}}'_i = \rho^{e \to v}\left(E'_i\right) \\
\mathbf{v}'_i = \phi^v\left(\overline{\mathbf{e}}'_i, \mathbf{v}_i, \mathbf{u}\right) & \overline{\mathbf{e}}' = \rho^{e \to u}\left(E'\right) \\
\mathbf{u}' = \phi^u\left(\overline{\mathbf{e}}', \overline{\mathbf{v}}', \mathbf{u}\right) & \overline{\mathbf{v}}' = \rho^{v \to u}\left(V'\right)
\end{array}
, \tag{3}
$$

---

**Algorithm 1** Training algorithm of CNFGNN on the server side.

---

**Server executes:**

1: Initialize server-side GN weights $\boldsymbol{\theta}_{GN}^{(0)}$, client model weights $\bar{\boldsymbol{\theta}}_c^{(0)} = \{\bar{\boldsymbol{\theta}}_c^{(0),enc}, \bar{\boldsymbol{\theta}}_c^{(0),dec}\}$.

2: **for** each node $i \in \mathcal{V}$ in parallel **do**

3:     Initialize client model $\boldsymbol{\theta}_{c,i}^{(0)} = \bar{\boldsymbol{\theta}}_c^{(0)}$.

4:     Initialize graph encoding on node $\boldsymbol{h}_{G,c,i} = \boldsymbol{h}_{G,c,i}^{(0)}$.

5: **end for**

6: **for** global round $r_g = 1, 2, \ldots, R_g$ **do**

7:     // **(1) Federated learning of on-node models.**

8:     **for** each client $i \in \mathcal{V}$ in parallel **do**

9:         $\boldsymbol{\theta}_{c,i} \leftarrow$ ClientUpdate($i$).

10:     **end for**

11:     $\bar{\boldsymbol{\theta}}_c \leftarrow \sum_{i \in \mathcal{V}} \frac{N_i}{N} \boldsymbol{\theta}_{c,i}$.

12:     **for** each client $i \in \mathcal{V}$ in parallel **do**

13:         Initialize client model: $\boldsymbol{\theta}_{c,i}^{(0)} = \bar{\boldsymbol{\theta}}_c$.

14:     **end for**

15:     // **(2) Temporal encoding update.**

16:     **for** each client $i \in \mathcal{V}$ in parallel **do**

17:         $\boldsymbol{h}_{c,i} \leftarrow$ ClientEncode($i$).

18:     **end for**

19:     // **(3) Split Learning of GN.**

20:     Initialize $\boldsymbol{\theta}_{GN}^{(r_g,0)} = \boldsymbol{\theta}_{GN}^{(r_g-1)}$.

21:     **for** server round $r_s = 1, 2, \ldots, R_s$ **do**

22:         $\{\boldsymbol{h}_{G,c,i} | i \in \mathcal{V}\} \leftarrow GN(\{\boldsymbol{h}_{c,i} | i \in \mathcal{V}\}; \boldsymbol{\theta}_{GN}^{(r_g,r_s-1)})$.

23:         **for** each client $i \in \mathcal{V}$ in parallel **do**

24:             $\nabla_{\boldsymbol{h}_{G,c,i}} \ell_i \leftarrow$ ClientBackward( $i, \boldsymbol{h}_{G,c,i}$).

25:             $\nabla_{\boldsymbol{\theta}_{GN}^{(r_g,r_s-1)}} \ell_i \leftarrow \boldsymbol{h}_{G,c,i}.\text{backward}(\nabla_{h_{G,c,i}} \ell_i)$.

26:         **end for**

27:         $\nabla_{\boldsymbol{\theta}_{GN}^{(r_g,r_s-1)}} \ell \leftarrow \sum_{i \in \mathcal{V}} \nabla_{\boldsymbol{\theta}_{GN}^{(r_g,r_s-1)}} \ell_i$.

28:         $\boldsymbol{\theta}_{GN}^{(r_g,r_s)} \leftarrow \boldsymbol{\theta}_{GN}^{(r_g,r_s-1)} - \eta_s \nabla_{\boldsymbol{\theta}_{GN}^{(r_g,r_s-1)}} \ell$.

29:     **end for**

30:     $\boldsymbol{\theta}_{GN}^{(r_g)} \leftarrow \boldsymbol{\theta}_{GN}^{(r_g,R_s)}$.

31:     // **(4) On-node graph embedding update.**

32:     $\{\boldsymbol{h}_{G,c,i} | i \in \mathcal{V}\} \leftarrow GN(\{\boldsymbol{h}_{c,i} | i \in \mathcal{V}\}; \boldsymbol{\theta}_{GN}^{(r_g)})$.

33:     **for** each client $i \in \mathcal{V}$ in parallel **do**

34:         Set graph encoding on client as $\boldsymbol{h}_{G,c,i}$.

35:     **end for**

36: **end for**

---

**Algorithm 2** Training algorithm of CNFGNN on the client side.

---

**ClientUpdate($i$):**

1: **for** client round $r_c = 1, 2, \ldots, R_c$ **do**

2:     $\boldsymbol{h}_{c,i}^{(r_c)} \leftarrow Encoder_i(\boldsymbol{x}_i; \boldsymbol{\theta}_{c,i}^{(r_c-1),enc})$.

3:     $\hat{\boldsymbol{y}}_i \leftarrow Decoder_i( x_{i,m}, [\boldsymbol{h}_{c,i}^{(r_c)}; \boldsymbol{h}_{G,c,i}]; \boldsymbol{\theta}_{c,i}^{(r_c-1),dec})$.

4:     $\ell_i \leftarrow \ell(\hat{\boldsymbol{y}}_i, \boldsymbol{y})$.

5:     $\boldsymbol{\theta}_{c,i}^{(r_c)} \leftarrow \boldsymbol{\theta}_{c,i}^{(r_c-1)} - \eta_c \nabla_{\boldsymbol{\theta}_{c,i}^{(r_c-1)}} \ell_i$.

6: **end for**

7: $\boldsymbol{\theta}_{c,i} = \boldsymbol{\theta}_{c,i}^{(R_c)}$.

8: **return** $\boldsymbol{\theta}_{c,i}$ to server.

**ClientEncode($i$):**

1: **return** $\boldsymbol{h}_{c,i} = Encoder_i(\boldsymbol{x}_i; \boldsymbol{\theta}_{c,i}^{enc})$ to server.

**ClientBackward($i, h_{G,c,i}$):**

1: $\hat{\boldsymbol{y}}_i \leftarrow Decoder_i(x_{i,m}, [h_{c,i}; h_{G,c,i}]; \boldsymbol{\theta}_{c,i}^{dec})$.

2: $\ell_i \leftarrow \ell(\hat{\boldsymbol{y}}_i, \boldsymbol{y})$.

3: **return** $\nabla_{\boldsymbol{h}_{G,c,i}} \ell_i$ to server.

---

where $\mathbf{e}_k, \mathbf{v}_i, \mathbf{u}$ are edge features, node features and global features respectively. $\phi^e, \phi^v, \phi^u$ are neural networks. $\rho^{e \to v}, \rho^{e \to u}, \rho^{v \to u}$ are aggregation functions such as summation. As shown in Figure 1b, we choose a 2-layer GN with residual connections for all experiments. We set $\mathbf{v}_i = \boldsymbol{h}_{c,i}$, $\mathbf{e}_k = W_{r_k,s_k}$ ($W$ is the adjacency matrix) , and assign the empty vector to $\mathbf{u}$ as the input of the first GN layer. The server-side GN outputs embeddings $\{\boldsymbol{h}_{G,c,i} \mid i \in \mathcal{V}\}$ for all nodes, and sends the embedding of each node correspondingly.

### 3.2.3 ALTERNATING TRAINING OF NODE-LEVEL AND SPATIAL MODELS

One challenge brought about by the cross-node federated learning requirement and the server-side GN model is the high communication cost in the training stage. Since we distribute different parts of the model on different devices, Split Learning proposed by (Singh et al., 2019) is a potential solution

Table 1: Statistics of datasets PEMS-BAY and METR-LA.

| Dataset | # Nodes | # Directed Edges | # Train Seq | # Val Seq | # Test Seq |
|---------|---------|------------------|-------------|-----------|------------|
| PEMS-BAY | 325 | 2369 | 36465 | 5209 | 10419 |
| METR-LA | 207 | 1515 | 23974 | 3425 | 6850 |

for training, where hidden vectors and gradients are communicated among devices. However, when we simply train the model end-to-end via Split Learning, the central server needs to receive hidden states from all nodes and to send node embeddings to all nodes in the forward propagation, then it must receive gradients of node embeddings from all nodes and send back gradients of hidden states to all nodes in the backward propagation. Assume all hidden states and node embeddings have the same size $S$, the total amount of data transmitted in each training round of the GN model is $4|\mathcal{V}|S$.

To alleviate the high communication cost in the training stage, we instead alternately train models on nodes and the GN model on the server. More specifically, in each round of training, we (1) fix the node embedding $\boldsymbol{h}_{G,c,i}$ and optimize the encoder-decoder model for $R_c$ rounds, then (2) we optimize the GN model while fixing all models on nodes. Since models on nodes are fixed, $\boldsymbol{h}_{c,i}$ stays constant during the training of the GN model, and the server only needs to fetch $\boldsymbol{h}_{c,i}$ from nodes before the training of GN starts and only to communicate node embeddings and gradients. Therefore, the average amount of data transmitted in each round for $Rs$ rounds of training of the GN model reduces to $\frac{2+2R_s}{R_s}|\mathcal{V}|S$. We provide more details of the training procedure in Algorithm 1 and Algorithm 2.

To more effectively extract temporal features from each node, we also train the encoder-decoder models on nodes with the FedAvg algorithm proposed in (McMahan et al., 2017). This enables all nodes to share the same feature extractor and thus share a joint hidden space of temporal features, which avoids the potential overfitting of models on nodes and demonstrates faster convergence and better prediction performance empirically.

## 4 EXPERIMENTS

We evaluate the performance of CNFGNN and all baseline methods on the traffic forecasting task, which is an important application for spatio-temporal data modeling. We reuse the following two real-world large-scale datasets in (Li et al., 2018) and follow the same preprocessing procedures: (1) **PEMS-BAY**: This dataset contains the traffic speed readings from 325 sensors in the Bay Area over 6 months from Jan 1st, 2017 to May 31st, 2017. (2) **METR-LA**: This dataset contains the traffic speed readings from 207 loop detectors installed on the highway of Los Angeles County over 4 months from Mar 1st, 2012 to Jun 30th, 2012.

For both datasets, we construct the adjacency matrix of sensors using the Gaussian kernel with a threshold: $W_{i,j} = d_{i,j}$ if $d_{i,j} >= \kappa$ else 0, where $d_{i,j} = \exp\left(-\frac{\text{dist}(v_i,v_j)^2}{\sigma^2}\right)$, $\text{dist}(v_i, v_j)$ is the road network distance from sensor $v_i$ to sensor $v_j$, $\sigma$ is the standard deviation of distances and $\kappa$ is the threshold. We set $\kappa = 0.1$ for both datasets.

We aggregate traffic speed readings in both datasets into 5-minute windows and truncate the whole sequence to multiple sequences with length 24. The forecasting task is to predict the traffic speed in the following 12 steps of each sequence given the first 12 steps. We show the statistics of both datasets in Table 1.

### 4.1 SPATIO-TEMPORAL DATA MODELING: TRAFFIC FLOW FORECASTING

**Baselines** We compare CNFGNN with the following baselines. (1) **GRU (centralized)**: a Gated Recurrent Unit (GRU) model trained with centralized sensor data. (2) **GRU + GN (centralized)**: a model directly combining GRU and GN trained with centralized data, whose architecture is similar to CNFGNN but all GRU modules on nodes always share the same weights. We see its performance as the upper bound of the performance of CNFGNN. (3) **GRU (local)**: for each node we train a GRU model with only the local data on it. (4) **GRU + FedAvg**: a GRU model trained with the Federated Averaging algorithm (McMahan et al., 2017). (5) **GRU + FMTL**: for each node we train

a GRU model using the federated multi-task learning (FMTL) with cluster regularization (Smith et al., 2017) given by the adjacency matrix. For each baseline, we have 2 variants of the GRU model to show the effect of on-device model complexity: one with 63K parameters and the other with 727K parameters. For CNFGNN, the encoder-decoder model on each node has 64K parameters and the GN model has 1M parameters.

Table 3: Comparison of the computation cost on edge devices and the communication cost. We use the amount of floating point operations (**FLOPS**) to measure the computational cost of models on edge devices. We also show the total size of data/parameters transmitted in the training stage (**Train Comm Cost**) until the model reaches its lowest validation error.

| Method | Comp Cost On Device (GFLOPS) | PEMS-BAY | | METR-LA | |
| --- | --- | --- | --- | --- | --- |
| | | RMSE | Train Comm Cost (GB) | RMSE | Train Comm Cost (GB) |
| GRU (63K) + FMTL | 0.159 | 3.961 | 57.823 | 11.548 | 99.201 |
| GRU (727K) + FMTL | 1.821 | 3.955 | 359.292 | 11.570 | 722.137 |
| CNFGNN (64K + 1M) | 0.162 | **3.822** | 237.654 | **11.487** | 222.246 |

**Discussion** Table 2 shows the comparison of forecasting performance and Table 3 shows the comparison of computation cost on device and communication cost of CNFGNN and baselines. We make the following observations. Firstly, when we compare the best forecasting performance of each baseline over the 2 GRU variants, GRU trained with FedAvg performs the worst in terms of forecasting performance compared to GRU trained with centralized data and GRU trained with local data (4.432 vs 4.010/4.124 on PEMS-BAY and 12.058 vs 11.730/11.801 on METR-LA), showing that the data distributions on different nodes are highly heterogeneous, and training one single model ignoring the heterogeneity is suboptimal.

Table 2: Comparison of performance on the traffic flow forecasting task. We use the Rooted Mean Squared Error (**RMSE**) to evaluate the forecasting performance.

| Method | PEMS-BAY | METR-LA |
| --- | --- | --- |
| GRU (centralized, 63K) | 4.124 | 11.730 |
| GRU (centralized, 727K) | 4.128 | 11.787 |
| GRU + GN (centralized, 64K + 1M) | **3.816** | **11.471** |
| GRU (local, 63K) | 4.010 | 11.801 |
| GRU (local, 727K) | 4.152 | 12.224 |
| GRU (63K) + FedAvg | 4.512 | 12.132 |
| GRU (727K) + FedAvg | 4.432 | 12.058 |
| GRU (63K) + FMTL | 3.961 | 11.548 |
| GRU (727K) + FMTL | 3.955 | 11.570 |
| CNFGNN (64K + 1M) | **3.822** | **11.487** |

Secondly, both the GRU+FMTL baseline and CNFGNN consider the spatial relations among nodes and show better forecasting performance than baselines without relation information. This shows that the modeling of spatial dependencies is critical for the forecasting task.

Lastly, CNFGNN achieves the lowest forecasting error on both datasets. The baselines that increases the complexity of on-device models (GRU (727K) + FMTL) gains slight or even no improvement at the cost of higher computation cost on edge devices and larger communication cost. However, due to its effective modeling of spatial dependencies in data, CNFGNN not only has the largest improvement of forecasting performance, but also keeps the computation cost on devices almost unchanged and maintains modest communication cost compared to baselines increasing the model complexity on devices.

## 4.2 INDUCTIVE LEARNING ON UNSEEN NODES

**Set-up** Another advantage of CNFGNN is that it can conduct inductive learning and generalize to larger graphs with nodes unobserved during the training stage. We evaluate the performance of CNFGNN under the following inductive learning setting: for each dataset, we first sort all sensors based on longitudes, then use the subgraph on the first $\eta\%$ of sensors to train the model and evaluate

Table 4: Inductive learning performance measured with rooted mean squared error (RMSE).

| Method | PEMS-BAY | | | METR-LA | | |
|---|---|---|---|---|---|---|
| | 25% | 50% | 75% | 25% | 50% | 75% |
| GRU (63K) + FedAvg | 4.863 | 4.847 | 4.859 | **11.993** | 12.104 | 12.014 |
| CNFGNN (64K + 1M) | **4.541** | **4.598** | **4.197** | 12.013 | **11.815** | **11.676** |

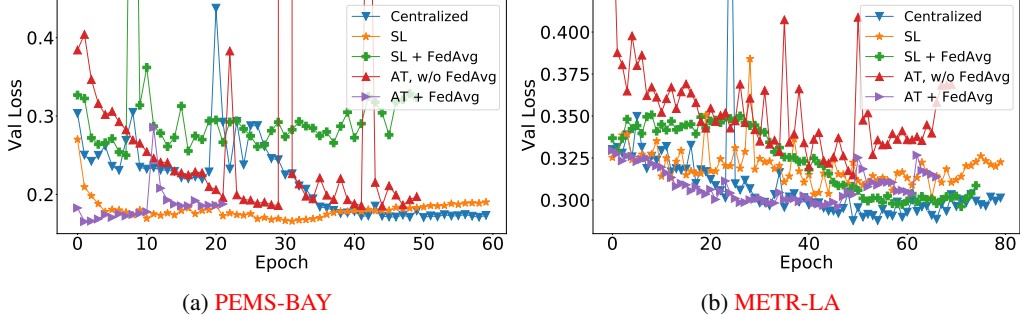

(a) PEMS-BAY            (b) METR-LA

Figure 2: Validation loss during the training stage of different training strategies.

the trained model on the entire graph. For each dataset we select $\eta\% = 25\%, 50\%, 75\%$. Over all baselines following the cross-node federated learning constraint, GRU (local) and GRU + FMTL requires training new models on unseen nodes and only GRU + FedAvg is applicable to the inductive learning setting.

**Discussion** Table 4 shows the performance of inductive learning of CNFGNN and GRU + FedAvg baseline on both datasets. We observe that under most settings, CNFGNN outperforms the GRU + FedAvg baseline (except on the METR-LA dataset with 25% nodes observed in training, where both models perform similarly), showing that CNFGNN has the stronger ability of generalization.

### 4.3 ABLATION STUDY: EFFECT OF ALTERNATING TRAINING AND FEDAVG ON NODE-LEVEL AND SPATIAL MODELS

**Baselines** We compare the effect of different training strategies of CN-FGNN: (1) **Centralized**: CNFGNN trained with centralized data where all nodes share one single encoder-decoder. (2) **Split Learning (SL)**: CNFGNN trained with split learning (Singh et al., 2019), where models on nodes and the model on the server are jointly trained by exchanging hidden vectors and gradients. (3) **Split**

Table 5: Comparison of test error (RMSE) and the communication cost during training of different training strategies of CNFGNN.

| Method | PEMS-BAY | | METR-LA | |
|---|---|---|---|---|
| | RMSE | Train Comm Cost (GB) | RMSE | Train Comm Cost (GB) |
| Centralized | **3.816** | - | **11.471** | - |
| SL | 3.914 | 350.366 | 12.186 | 307.627 |
| SL + FedAvg | 4.383 | 80.200 | 11.631 | 343.031 |
| AT, w/o FedAvg | 4.003 | 5221.576 | 11.912 | 2434.985 |
| AT + FedAvg | **3.822** | 237.654 | **11.487** | 222.246 |

**Learning + FedAvg (SL + FedAvg)**: A variant of SL that synchronizes the weights of encoder-decoder modules periodically with FedAvg. (4) Alternating training without Federated Averaging of models on nodes (**AT, w/o FedAvg**). (5) Alternating training with Federated Averaging on nodes described in Section 3.2.3 (**AT + FedAvg**).

**Discussion** Figure 2 shows the validation loss during training of different training strategies on PEMS-BAY and METR-LA datasets, and Table 5 shows their prediction performance and the com-

munication cost in training. We notice that (1) SL suffers from suboptimal prediction performance and high communication costs on both datasets; SL + FedAvg does not have consistent results on both datasets and its performance is always inferior to AT + FedAvg. AT + FedAvg consistently outperforms other baselines on both datasets, including its variant without FedAvg. (2) AT + FedAvg has the lowest communication cost on METR-LA and the 2nd lowest communication cost on PEMS-BAY, on which the baseline with the lowest communication cost (SL + FedAvg) has a much higher prediction error (4.383 vs 3.822). Both illustrate that our proposed training strategy, SL + FedAvg, achieves the best prediction performance as well as low communication cost compared to other baseline strategies.

### 4.4 ABLATION STUDY: EFFECT OF CLIENT ROUNDS AND SERVER ROUNDS

**Set-up**  We further investigate the effect of different compositions of the number of client rounds ($R_s$) in Algorithm 2 and the number of server rounds ($R_c$) in Algorithm 1. To this end, we vary both $R_c$ and $R_s$ over [1,10,20].

**Discussion**  Figure 3 shows the forecasting performance (measured with RMSE) and the total communication cost in the training of CN-FGNN under all compositions of ($R_c$, $R_s$) on the METR-LA dataset. We observe that: (1) Models with lower $R_c/R_s$ ratios ($R_c/R_s < 0.5$) tend to have lower forecasting errors while models with higher $R_c/R_s$ ratios ($R_c/R_s > 2$) have lower communication cost in training. This is because the lower ratio of $R_c/R_s$ encourages more frequent exchange of node information at the expense of higher communication cost, while the higher ratio of $R_c/R_s$ acts in the opposite way. (2) Models with similar $R_c/R_s$ ratios have similar communication costs, while those with lower

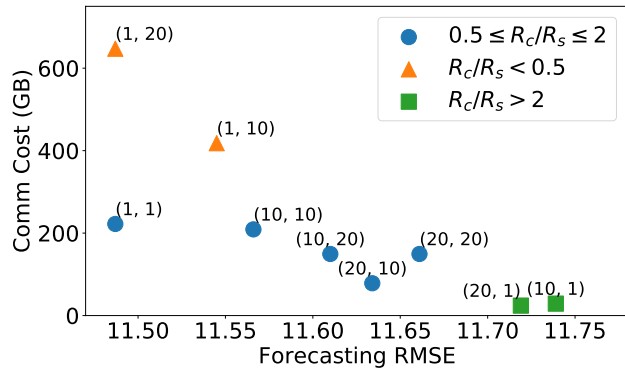

Figure 3: Effect of client rounds and server rounds ($R_c$, $R_s$) on forecasting performance and communication cost.

$R_c$ values perform better, corroborating our observation in (1) that frequent node information exchange improves the forecasting performance.

## 5 CONCLUSION

We propose Cross-Node Federated Graph Neural Network (CNFGNN), which bridges the gap between modeling complex spatio-temporal data and decentralized data processing by enabling the use of graph neural networks (GNNs) in the federated learning setting. We accomplish this by decoupling the learning of local temporal models and the server-side spatial model using alternating optimization of spatial and temporal modules based on split learning and federated averaging. Our experimental results on traffic flow prediction on two real-world datasets show superior performance as compared to competing techniques. Our future work includes applying existing GNN models with sampling strategies and integrating them into CNFGNN for large-scale graphs, extending CN-FGNN to a fully decentralized framework, and incorporating existing privacy-preserving methods for graph learning to CNFGNN, to enhance federated learning of spatio-temporal dynamics.

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

# A  APPENDIX

## A.1  DETAILED EXPERIMENT SETTINGS

Unless noted otherwise, all models are optimized using the Adam optimizer with the learning rate 1e-3.

**GRU (centralized)**  : Gated Recurrent Unit (GRU) model trained with centralized sensor data. The GRU model with 63K parameters is a 1-layer GRU with hidden dimension 100, and the GRU model with 727K parameters is a 2-layer GRU with hidden dimension 200.

**GRU (local)**  We train one GRU model for each node with the local data only.

**GRU + FedAvg**  We train a single GRU model with Federated Averaging (McMahan et al., 2017). We select 1 as the number of local epochs.

**GRU + FMTL**  We train one GRU model for each node using the federated multi-task learning (FMTL) with cluster regularization (Smith et al., 2017) given by the adjacency matrix. More specifically, the cluster regularization (without the L2-norm regularization term) takes the following form:

$$\mathcal{R}(\boldsymbol{W}, \boldsymbol{\Omega}) = \lambda \mathrm{tr}(\boldsymbol{W}\boldsymbol{\Omega}\boldsymbol{W}^T). \tag{A1}$$

Given the constructed adjacency matrix $\boldsymbol{A}$, $\boldsymbol{\Omega} = \frac{1}{|\mathcal{V}|}(\boldsymbol{D} - \boldsymbol{A}) = \frac{1}{|\mathcal{V}|}\boldsymbol{L}$, where $\boldsymbol{D}$ is the degree matrix and $\boldsymbol{L}$ is the Laplacian matrix. Equation A1 can be reformulated as:

$$
\begin{aligned}
\mathcal{R}(\boldsymbol{W}, \boldsymbol{\Omega}) = \lambda \mathrm{tr}(\boldsymbol{W}\boldsymbol{\Omega}\boldsymbol{W}^T) &= \frac{\lambda}{|\mathcal{V}|}\mathrm{tr}(\boldsymbol{W}\boldsymbol{L}\boldsymbol{W}^T) \\
&= \frac{\lambda}{|\mathcal{V}|}\mathrm{tr}(\sum_{i \in \mathcal{V}} \boldsymbol{w}_i \sum_{j \neq i} a_{ij}\boldsymbol{w}_i^T - \sum_{j \neq i} \boldsymbol{w}_i a_{ij}\boldsymbol{w}_j^T) \\
&= \lambda_1(\sum_{i \in \mathcal{V}} \sum_{j \neq i} \alpha_{i,j}\langle \boldsymbol{w}_i, \boldsymbol{w}_i - \boldsymbol{w}_j \rangle).
\end{aligned}
\tag{A2}
$$

We implement the cluster regularization via sharing model weights between each pair of nodes connected by an edge and select $\lambda_1 = 0.1$.

**CNFGNN**  We use a GRU-based encoder-decoder model as the model on nodes, which has 1 GRU layer and hidden dimension 64. We use a 2-layer Graph Network (GN) with residual connections as the Graph Neural Network model on the server side. We use the same network architecture for the edge/node/global update function in each GN layer: a multi-layer perceptron (MLP) with 3 hidden layers, whose sizes are [256, 256, 128] respectively. We choose $R_c = 1, R_s = 20$ for experiments on PEMS-BAY, and $R_c = 1, R_s = 1$ for METR-LA.

## A.2  CALCULATION OF COMMUNICATION COST

We denote $R$ as the number of communication rounds for one model to reach the lowest validation error in the training stage.

**GRU + FMTL**  Using Equation A2, in each communication round, each pair of nodes exchange their model weights, thus the total communicated data amount is calculated as:

$$R \times \#\text{nonself directed edges} \times \text{size of node model weights}. \tag{A3}$$

**CNFGNN (AT + FedAvg)**  In each communication round, the central server fetches and sends back model weights to each node for Federated Averaging, and transmits hidden vectors and gradients for Split Learning. The total communicated data amount is calculated as:

$$
\begin{aligned}
R \times (\#\text{nodes} \times \text{size of node model weights} \times 2 \\
+ (1 + 2 * \text{server round} + 1) \times \#\text{nodes} \times \text{hidden state size}).
\end{aligned}
\tag{A4}
$$

**CNFGNN (SL)**   In each communication round, each node sends and fetches hidden vectors and graidents twice (one for encoder, the other for decoder) and the total communicated data amount is:

$$R \times 2 \times 2 \times \#\text{nodes} \times \text{hidden state size}. \tag{A5}$$

**CNFGNN (SL + FedAvg)**   Compared to CNFGNN (SL), the method has extra communcation cost for FedAvg in each round, thus the total communicated data amount is:

$$R \times (\#\text{nodes} \times \text{size of node model weights} \times 2 + 2 \times 2 \times \#\text{nodes} \times \text{hidden state size}). \tag{A6}$$

**CNFGNN (AT, w/o FedAvg)**   Compared to CNFGNN (AT + FedAvg), there is no communcation cost for the FedAvg part, thus the total communcated data amount is:

$$R \times (1 + 2 * \text{server round} + 1) \times \#\text{nodes} \times \text{hidden state size}. \tag{A7}$$

Table A1: Parameters used for calculating the communication cost of GRU + FMTL.

| | Method | GRU (63K) + FMTL | GRU (727K) + FMTL |
|---|---|---|---|
| | Node Model Weights Size (GB) | 2.347E-4 | 2.708E-3 |
| PEMS-BAY | #Nonself Directed Edges | 2369 | |
| | R | 104 | 56 |
| | Train Comm Cost (GB) | 57.823 | 359.292 |
| METR-LA | #Nonself Directed Edges | 1515 | |
| | R | 279 | 176 |
| | Train Comm Cost (GB) | 99.201 | 722.137 |

Table A2: Parameters used for calculating the communication cost of CNFGNN (AT + FedAvg).

| Node Model Weights Size (GB) | | 2.384E-4 |
|---|---|---|
| PEMS-BAY | #Nodes | 325 |
| | Hidden State Size (GB) | 2.173E-3 |
| | Server Round | 20 |
| | R | 2 |
| | Train Comm Cost (GB) | 237.654 |
| METR-LA | #Nodes | 207 |
| | Hidden State Size (GB) | 1.429E-3 |
| | Server Round | 1 |
| | R | 46 |
| | Train Comm Cost (GB) | 222.246 |

## A.3   INDUCTIVE LEARNING

We have added results using 90% and 5% data on both datasets and we show the table of inductive learning results as Table A6. We observe that: (1) With the portion of visible nodes in the training stage increasing, the prediction error of CNFGNN decreases drastically. However, the increase of the portion of visible nodes has negligible contribution to the performance of GRU + FedAvg after the portion surpasses 25%. Since increasing the ratio of seen nodes in training introduces more complex relationships among nodes to the training data, the difference of performance illustrates that CNFGNN has a stronger capability of capturing complex spatial relationships. (2) When the ratio of

Table A3: Parameters used for calculating the communication cost of CNFGNN (SL).

| | | |
|---|---|---|
| PEMS-BAY | #Nodes | 325 |
| | Hidden State Size (GB) | 2.173E-3 |
| | R | 31 |
| | Train Comm Cost (GB) | 350.366 |
| METR-LA | #Nodes | 207 |
| | Hidden State Size (GB) | 1.429E-3 |
| | R | 65 |
| | Train Comm Cost (GB) | 307.627 |

Table A4: Parameters used for calculating the communication cost of CNFGNN (SL + FedAvg).

| Node Model Weights Size (GB) | 2.384E-4 | |
|---|---|---|
| PEMS-BAY | #Nodes | 325 |
| | Hidden State Size (GB) | 2.173E-3 |
| | R | 7 |
| | Train Comm Cost (GB) | 80.200 |
| METR-LA | #Nodes | 207 |
| | Hidden State Size (GB) | 1.429E-3 |
| | R | 71 |
| | Train Comm Cost (GB) | 343.031 |

Table A5: Parameters used for calculating the communication cost of CNFGNN (AT, w/o FedAvg).

| | | |
|---|---|---|
| PEMS-BAY | #Nodes | 325 |
| | Hidden State Size (GB) | 2.173E-3 |
| | Server Round | 20 |
| | R | 44 |
| | Train Comm Cost (GB) | 5221.576 |
| METR-LA | #Nodes | 207 |
| | Hidden State Size (GB) | 1.429E-3 |
| | Server Round | 1 |
| | R | 49 |
| | Train Comm Cost (GB) | 2434.985 |

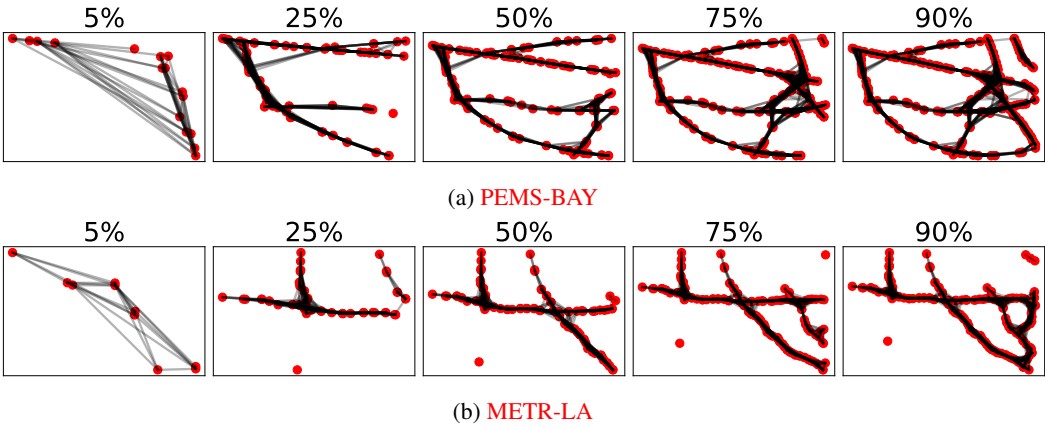

(a) PEMS-BAY

(b) METR-LA

Figure A1: Visualization of subgraphs visible in training under different ratios.

Table A6: Inductive learning performance measured with rooted mean squared error (RMSE).

| Method | PEMS-BAY | | | | | METR-LA | | | | |
|---|---|---|---|---|---|---|---|---|---|---|
| | 5% | 25% | 50% | 75% | 90% | 5% | 25% | 50% | 75% | 90% |
| GRU (63K) + FedAvg | **5.087** | 4.863 | 4.847 | 4.859 | 4.866 | **12.128** | **11.993** | 12.104 | 12.014 | 12.016 |
| CNFGNN (64K + 1M) | 5.869 | **4.541** | **4.598** | **4.197** | **3.942** | 13.931 | 12.013 | **11.815** | **11.676** | **11.629** |

visible nodes in training is extremely low (5%), there is not enough spatial relationship information in the training data to train the GN module in CNFGNN, and the performance of CNFGNN may not be ideal. We visualize the subgraphs visible in training under different ratios in Figure A1. However, as long as the training data covers a moderate portion of the spatial information of the whole graph, CNFGNN can still leverage the learned spatial connections among nodes effectively and outperforms GRU+FedAvg. We empirically show that the necessary ratio can vary for different datasets (25% for PEMS-BAY and 50% for METR-LA).

## A.4    THE HISTOGRAMS OF DATA ON DIFFERENT NODES

We show the histograms of traffic speed on different nodes of PEMS-BAY and METR-LA in Figure A2. For each dataset, we only show the first 100 nodes ranked by their IDs for simplicity. The histograms show that the data distribution varies with nodes, thus data on different nodes are not independent and identically distributed.

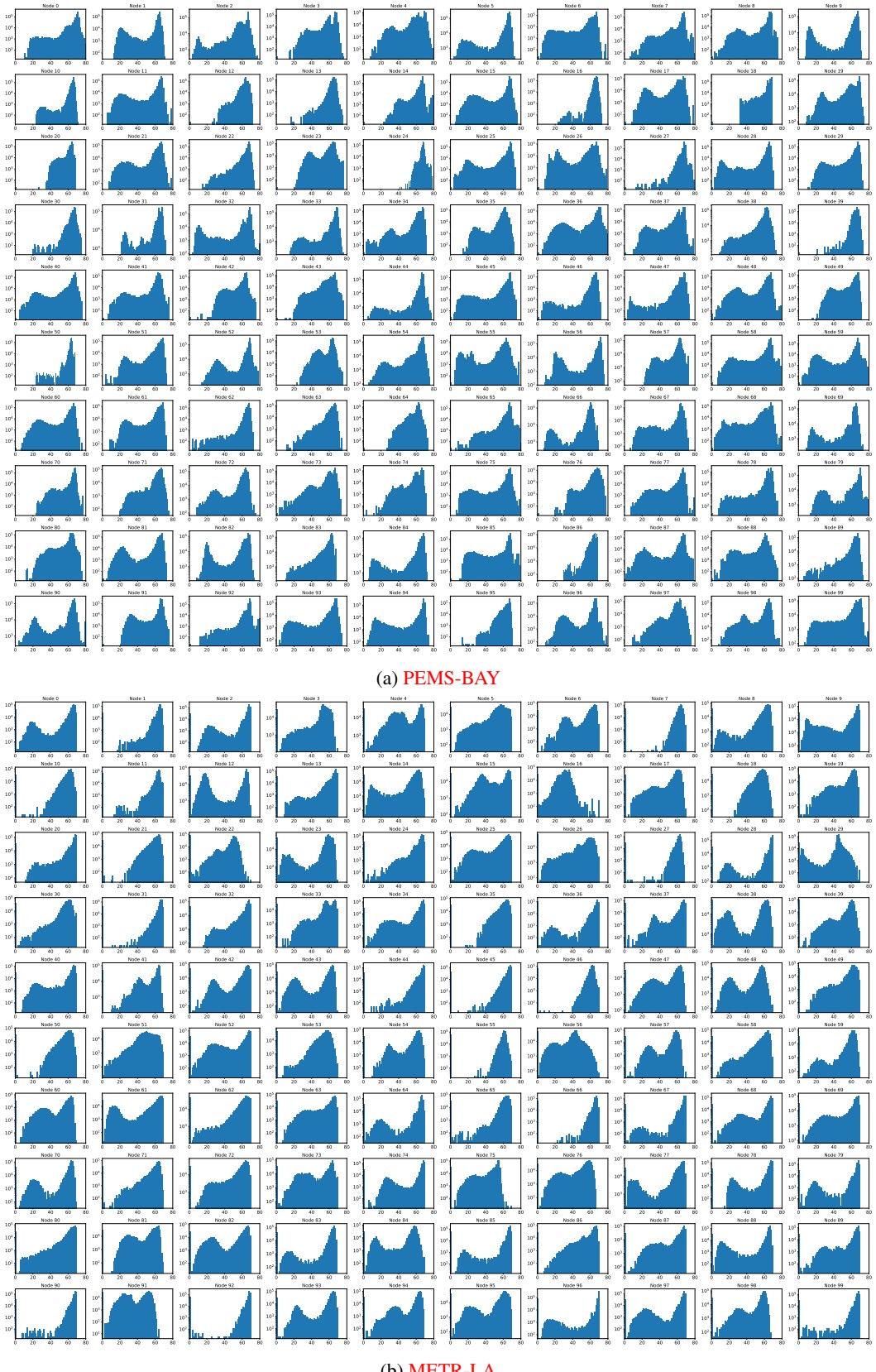

(a) PEMS-BAY

(b) METR-LA

Figure A2: The histograms of data on the first 100 nodes ranked by ID.

