# OpenReview forum: "Cross-Node Federated Graph Neural Network for Spatio-Temporal Data Modeling"
_ICLR.cc/2021/Conference — Reject_

### Official Review · AnonReviewer2 · 2020-10-27
**This paper extends federated learning to graph-based spatio-temporal forecasting tasks. However, privacy issues, communication cost issues, and experiments need to be improved and clarified.**

**Rating:** 5
**Confidence:** 3

**Review:**

This paper presents an algorithm to model the complex spatio-temporal dependencies across multiple nodes under a federated learning setting. Specifically, the authors employ a GRU-based encoder-decoder architecture to construct the node-level temporal dynamics on each node and utilize a GCN to capture server-level spatial dynamics on the server.

The model consists of four major steps. First, each node encodes the local data and sends the extracted temporal vectors to the server. The server then collects all temporal vectors, builds the graph, extracts the spatial vectors, and broadcasts the server-level dynamics. Next, each node combines the node-level temporal dynamics and the server-level spatial dynamics to decode the prediction.  Finally, the backpropagation algorithm is performed.

Pros -------------

- It is interesting to extend the concept of federated learning to graph-based spatio-temporal forecasting tasks.
- The paper is well-written and structured.
- Split learning seems to be a good idea to reduce communication costs.

Cons  -------------

- Privacy concern: This paper introduces some privacy-preserving methods in Related Work, but the corresponding protection methods do not present in the paper. Or the author regards the extracted feature vectors and corresponding gradients are not sensitive.

- Communication costs: Although the model makes several adjustments, the amount of data transmitted in a complete training phase is still very large (200+GB), which places a demanding requirement on the capability of edge devices. In addition, I am curious about where communication costs mainly come from. the vectors and gradients from SplitNN? or the model parameters from FedAvg?

- Experiments:  An important baseline is missing, a model that aggregates all the data and directly combines GRU with GNN to predict the traffic flow without a federated setting (like DCRNN). In my view, this model should be the upper bound of the model performance, just like what the author did in section 4.3-(1). Besides, there is no explanation for the little change in the performance of GRU+FedAvg in section 4.2, what if we use more data like 90% or fewer data like 5%. Besides, the main difference between GRU+FedAvg and CNFGNN is whether to use GNN to capture spatial dependencies.  Is it spatial dependencies as the key reason that achieves better performances in an inductive learning setting?

---

> ### Author Response · Authors · 2020-11-21
> **Response to AnonReviewer2**
>
> Thank you for your comments and suggestions to improve the paper. Below are our responses to your comments:
>
> 1. **Privacy concern**
>
> > The focus of our paper is to design and develop a model to leverage the spatio-temporal relationships under the federated learning constraint, where the data is accessible in a decentralized way. To the best of our knowledge, our work is a first work in this direction.
>
> > Although we do not explicitly address the privacy aspect [1,2] in this work, various existing privacy-preserving techniques, such as Differential Privacy (DP), Secure Multi-Party Computation (MPC) and Homomorphic Encryption (HE) can potentially be integrated into our proposed framework and help build privacy-preserving federated learning systems. Extending our framework with existing privacy-preserving methods constitutes an interesting future work towards a fully practical federated learning system, as discussed in Section 2 and Section 5.
>
> 2. **Communication costs**
>
> > 2.1. We use alternating training to reduce the communication cost induced by Split Learning, as we have discussed in Section 3.2.3. Compared to methods increasing the complexity of on-device models, our proposed method maintains modest communication cost while gains more improvement of prediction performance.
>
> > 2.2 The communication cost mainly comes from the vectors and gradients transmitted among nodes. In equation (A4), $(1 + 2 * \mathrm{server\ round} + 1) \times \mathrm{nodes} \times \mathrm{hidden\ state\ size}$ is the amount of communicated data of vectors and gradients in each round of training, and we can calculate its ratio over the total amount of communicated data is 99.87% (PEMS-BAY) and 97.99% (METR-LA).
>
> 3. **Experiments**
>
> > 3.1. We have added the results of the model directly combining GRU with GNN trained on centralized data in **Table 2** and **Table 3**. Results match our expectation that the performance of the model trained on centralized data is the performance upper bound.
>
> > 3.2. We have added results using 90% and 5% data on both datasets. We visualize the subgraph under each ratio in Figure A1, and we show the table of inductive learning results here. We observe that our proposed method CNFGNN achieves better performance than the FedAvg baseline when trained on data with adequate spatial coverage.
>
> > **PEMS-BAY**:
>
> > | Method | 5% | 25% | 50% | 75% | 90% |
> |:-----:|:-----:|:-----:|:-----:|:-----:|:-----:|
> | GRU + FedAvg |  **5.087** |	4.863|	4.847|	4.859|	4.866|
> | CNFGNN | 5.869| **4.541**|**4.598**|**4.197**|**3.942**|
>
> > **METR-LA**:
>
> > | Method | 5% | 25% | 50% | 75% | 90% |
> |:-----:|:-----:|:-----:|:-----:|:-----:|:-----:|
> | GRU + FedAvg | **12.128**|**11.993**|	12.104|	12.014|	12.016|
> | CNFGNN | 13.931|	12.013|	**11.815**|	**11.676**|	**11.629**|
>
> > From the above tables we observe that:
>
> > (1) **CNFGNN captures complex spatial relationships**. With the portion of visible nodes in the training stage increasing, the prediction error of CNFGNN decreases drastically. However, the increase of the portion of visible nodes has negligible contribution to the performance of GRU + FedAvg after the portion surpasses 25%. Since increasing the ratio of seen nodes in training introduces more complex relationships among nodes to the training data, the difference of performance illustrates that CNFGNN has a stronger capability of capturing complex spatial relationships.
>
> > (2) **Adequate spatial coverage in training data is necessary for CNFGNN**. When the ratio of visible nodes in training is extremely low (5%), there is not enough spatial relationship information in the training data to train the GN module in CNFGNN, and the performance of CNFGNN may not be ideal. We visualize the subgraphs visible in training under different ratios in Figure  A1.  However, as long as the training data covers a moderate portion of the spatial information of the whole graph, CNFGNN can still leverage the learned spatial connections among nodes effectively and outperforms GRU+FedAvg. We empirically show that the necessary ratio can vary for different datasets (25% for PEMS-BAY and 50% for METR-LA).
>
> > 3.3. We think the capture of spatial dependencies is the key reason for CNFGNN to achieve better performance in an inductive learning setting. In our reply 3.2, we show that with the increase of information of spatial dependencies, CNFGNN rapidly benefits from it, while GRU+FedAvg cannot utilize the spatial dependencies effectively and quickly fall behind in terms of inductive learning performance. In Table 2 we also show that both the GRU+FMTL baseline and CNFGNN consider the spatial relations among nodes and show better forecasting performance than baselines without relation information. However, FMTL suffers from the limitation of regularization based methods due to the assumption that graphs only encode similarity of nodes [1] and cannot operate in inductive learning settings where only a fraction of devices are observed during training.

---

> > ### Author Response · Authors · 2020-11-21
> > **Response to AnonReviewer2 (Part 2)**
> >
> > References:
> >
> > [1]. Zhu, Ligeng, Zhijian Liu, and Song Han. "Deep leakage from gradients." Advances in Neural Information Processing Systems. 2019.
> >
> > [2]. Rigaki, Maria, and Sebastian Garcia. "A survey of privacy attacks in machine learning." arXiv preprint arXiv:2007.07646 (2020).
> >
> > [3]. Thomas N. Kipf and Max Welling. Semi-supervised classification with graph convolutional networks. In International Conference on Learning Representations (ICLR), 2017.

---

### Official Review · AnonReviewer4 · 2020-10-28
**Network modeling using Cross-Node-Federated-Graph-Neural-Network (CNFGNN)**

**Rating:** 6
**Confidence:** 2

**Review:**

The paper proposes a new modeling framework which would utilize the spatio-temporal information in the data by developing a Cross-Node-Federated-Graph-Neural-Network (CNFGNN). The methodology seems to be clearly explained while it relies on many existing methods in the literature which limits the novelty of the paper. The model is applied to two real data sets and outperforms some existing methods. Overall, I find the methodology interesting.

Some comments:

In the traffic flow forecasting data: (1) Have you tried different weights (W_{i,j}) instead of the Gaussian kernel function? Does the performance measurements change by using a different weight function? It would be appropriate to show the robustness of the method with respect to changes in the spatial weight selection. (2) Removing trend and seasonality is a common practice in time series forecasting, but it is not clear whether this step is addressed properly or not. What about the seasonality in the data? Is this issue resolved in the preprocessing?

---

> ### Author Response · Authors · 2020-11-18
> **Response to AnonReviewer4**
>
> Thank you for your comments and suggestions to improve the paper. Below are our responses to the main points of your comments:
>
> 1. *“Have you tried different weights (W_{i,j}) instead of the Gaussian kernel function?”*
>
> > The focus of our paper is on how to design and train a model to leverage spatio-temporal relationships **under the federated learning constraint**. To this end, we construct the graph using the Gaussian kernel function, which is a popular choice for Traffic forecasting applications [1-4]. Indeed any improvements on the spatio-temporal data modeling aspect such as learnable spatial relationships [5,6] or other ways to choose the weights can be leveraged in conjunction with the proposed model to further improve the prediction performance, and offer promising future extensions of our work.
>
> 2. *“Removing trend and seasonality is a common practice in time series forecasting, but it is not clear whether this step is addressed properly or not.”*
>
> > We choose RNN based models for encoding temporal information on nodes due to their strong abilities to model complex patterns in time series. We follow the same preprocessing steps for traffic data widely used in related works [1-6], where results show that RNN based methods outperform other baselines without special preprocessing w.r.t trend and seasonality.
>
>
> References:
>
> [1]. Yaguang Li, Rose Yu, Cyrus Shahabi, and Yan Liu. Diffusion convolutional recurrent neural network: Data-driven traffic forecasting. In International Conference on Learning Representations (ICLR ’18), 2018.
>
> [2]. Bing Yu, Haoteng Yin, and Zhanxing Zhu. Spatio-Temporal Graph Convolutional Networks: A Deep Learning Framework for Traffic Forecasting. In Proc. of IJCAI, 2018.
>
> [3]. Zonghan Wu, Shirui Pan, Guodong Long, Jing Jiang, and Chengqi Zhang. Graph WaveNet for Deep Spatial-Temporal Graph Modeling. In Proc. of IJCAI, 2019.
>
> [4]. Weiqi Chen, Ling Chen, Yu Xie, Wei Cao, Yusong Gao, and Xiaojie Feng. Multi-Range Attentive Bicomponent Graph Convolutional Network for Traffic Forecasting. In Proc. of AAAI, 2019.
>
> [5]. Chuanpan Zheng, Xiaoliang Fan, Cheng Wang, and Jianzhong Qi. GMAN: A Graph Multi-Attention Network for Traffic Prediction. In Proc. of AAAI, 2020.
>
> [6]. Wu, Zonghan, et al. Connecting the Dots: Multivariate Time Series Forecasting with Graph Neural Networks. In Proc. of KDD, 2020.

---

### Official Review · AnonReviewer3 · 2020-10-29
**To broaden the application of federated learning, the author proposed a node-level distributed training framework for an existing spatial-temporal model (GCN + GRU), which aims to protect the privacy or circumvent the difficulty of centralizing datasets.**

**Rating:** 3
**Confidence:** 5

**Review:**

The proposed framework is demonstrated with two datasets from the transportation sensor
network. The experimental results on model performance and training property demonstrate the
efficacy of the proposed method empirically. The model used is from existing works, but the
training framework is relatively novel in the FL community.
Strengths
1. It is very encouraging to support advanced models for federated learning. The author
goes in the right direction. As far as I know, this is the first time that GCN+GRU is
discussed under the context of federated learning.
2. The techniques discussed in this work is comprehensive since the authors introduce
both the model and the training method.
Weakness
***1. Motivation is not well-discussed***
(1) From the introduction section, the author claim that the node-level datasets cannot be
centralized. But in practice, at least sensors in IoT networks CAN be centralized to edge
servers.
(2) Especially, the datasets used in experiments does not have any privacy issue. It is very
strange to me why we should concern the privacy of the *traffic speed* from road network
sensors. We can check the road network speed in real-time using Google Map. In other words,
the speed of a specific sensor is not sensitive and can be acquired by public service. Currently,
the Google Map example demonstrates that the road network sensors can be uploaded to
servers from transportation agencies without any privacy concerns.
(3) Another serious problem is that the DNN-based model (encoder-decoder used in the
proposed method) are not deployable in the sensor device due to computational resource
constraint. In practice, the number of road network sensor in transportation is huge, so it is
impractical to ask for an expensive upgrading which requires to install a powerful DNN enabled
chips.
***2. No technical contribution***
The focus of this work is diverse (both model and training), but lack of contribution in each
aspect.
(1) The authors argue the contribution of modeling (e.g., 4.1 modeling spatial dependencies, 4.2
“inductive v.s. transductive”). However, for the modeling part, I believe the contribution is from
GCN+GRU-based modeling, which is already there in many data mining publications (Equation
(1)(2)(3) are not newly proposed models). Please refer to [1][2]. The model used in this paper is
a simplified version of these two GCN+RNN-based models in transportation. More advanced
models following these two can obtain better performance than the model used in this paper.
[1] Spatio-Temporal Graph Convolutional Networks: A Deep Learning Framework for Traffic
Forecasting
[2] Diffusion convolutional recurrent neural network: Data-driven traffic forecasting
(2) The authors also claim the benefit of alternating optimization (4.3, 4.4). It is good to know
such an alternating method works well in practice. However, the author does not provide any
in-depth analysis or comprehensive experimental evidence (see my comments 3 below for
details).
In short, I suggest the author focuses on one aspect of federated learning to consolidate the
contribution. At least, the current version raises my concern that this work is a simple
combination of many techniques without meta-contribution in each one. If the overall
characteristics are good, it should be encouraged. However, the effectiveness is not obvious:
higher accuracy, security, fairness, privacy, communication, and computation efficiency, I
cannot see any of these advantages of such a combination:
[Model Performance] In terms of higher accuracy, as mentioned in (2), the model used is not
novel. More advanced models can obtain higher accuracy. See following works of 2-(2) [1][2] for
details.
[Optimization] The experimental results of the alternating optimization has some
counter-intuition results. See 3 for details.
[Security/Privacy] As for security and privacy, the hidden vector exchange may leak privacy but
this work does not discuss it. See 8 for details.
[Computational Efficiency] The model used in edge devices are too computational expensive for
resource constrained sensors. See 4 for details.
***3. The training Algorithm is novel but the experimental result is not convincing***
The training method is relatively novel in federated learning. However, the authors should either
provide enough intuition to explain why this work (any related works in optimization theory?) or
demonstrate it with experimental design. The author attempts to prove that alternating training
works better but it’s not convincing. In Section 4.3, all the experimental results are not
convincing or counterintuitive:
First, I am surprised that the authors have a result to say that split learning performs worse than
centralized training. In the essence of optimization, split learning is the same as centralized
training. The only difference is that split learning offloads part of the model architecture to the
server-side to reduce the communication/computation cost.
Second, due to the first reason, that “AT w/o FedAvg” and “AT + FedAvg” works better than SL
can NOT demonstrate the effectiveness of alternating optimization. As I mentioned previously,
to prove the effectiveness of alternating optimization, we need more analysis or comprehensive
experimental design.
Third, another counterintuitive result is that AT + FedAvg performs better than centralized
training (the red curve is lower than the blue curve). Normally, in federated learning, one should
believe that centralized training is the lower bound of training/validation errors. An alternating
local SGD method normally leads to variance bias in the training process, thus I believe it’s
impossible to obtain a better performance than centralized training.
The last concern is that the experiments only perform at one dataset (METR-LA). Please also
check whether the same conclusion holds in another dataset.
***4. ​Communication and computational cost***
The computational cost is evaluated, but what’s the implication of the number showing in terms
of FLOPS? I don’t think the resource constrained road network sensor can handle so many
FLOPS. Does any running time result in a real-world sensor device? If testing the sensor is
prohibited, please provide the running time result at a low-performance smartphone or other IoT
devices (NVIDIA edge GPU devices).
***5. ​Overall training time***
The overall training time and the bandwidth of exchanging hidden vectors should be mentioned.
If the network size of the exchange information is too high and the bandwidth is limited in
wireless communication (e.g., road sensor networks), a single iteration will cost too much time
or failure due to communication protocol constraint (e.g., in IoT setting, maximum payload
length is only 65535 bytes), which will lead to a long training time or impractical assumption.
Please discuss the size of exchange information (hidden vector/gradient) and the total training
time in revision.
***6. Scalability issue is ignored***
The proposed method does not use a client sampling strategy, a common practice in
cross-device FL (see the original FedAvg paper [1]), to mitigate the scalability issue. What the
performance if we want to learn 10 thousand sensors? Especially in the transportation setting,
the scalability is very important. However, the authors do not discuss this.
[1] Communication-Efficient Learning of Deep Networks from Decentralized Data.
https://arxiv.org/abs/1602.05629
***7. Dataset: Non-I.I.D. is not properly discussed***
A key assumption of Federated Learning is that datasets across devices are non-I.I.D. This is
largely ignored by the proposed methods. Slightly mentioning the heterogeneous property is not
enough. Please discuss the details of this in revision (e.g., show the distribution diagram). More
significantly, DNN models like encoder-decoder architectures normally eat a lot of samples, but
the number of samples in each node is small in practice. The assumption that each node has
enough dataset to train a good model is too strong (we cannot assume the edge device has the
storage capability to store months of time series data since the storage ability is limited at the
edge, ***4-6 months*** as the author mentioned). Maybe the alternating optimization method
helps, but it lacks analysis. In my opinion, it’s more interesting and practical to construct a
graph-level federated learning method since multiple nodes can be centralized to an edge
server belongs to a company or an agency like the transportation scenario.
***8. ​Privacy***
The authors discuss some privacy-preserving methods in related works, but the proposed
method does not include any privacy design. For example, [1] analyze potential leakage from
the hidden vectors, which proves that the proposed method does not have privacy advantages
than exchanging gradient/models.
[1] Cronus: Robust and Heterogeneous Collaborative Learning with Black-Box Knowledge
Transfer. https://arxiv.org/abs/1912.11279
***9. Lack of Related works***
Since alternating optimization (training algorithm) is the novel part of the proposed method,
there is a lack of literature review of this method, either in the general ML optimization literature,
or the specific federated learning publications. Without such a comparison, the novelty is not
convincing.
***10. Some misunderstandings of federated learning***
(1) In the first paragraph of the introduction, the authors claim that decentralized training can
improve latency, but this argument is vague. Improving training latency or inference latency?
Federated learning will cost time to train on devices, which requires a long time due to many
rounds of communication synchronization. For the inference, to improve the latency, we need
model compression techniques, which is also not the main goal of the proposed method.
(2) The term “decentralized training” is misused. In federated learning, decentralized training
also refers to training using a decentralized topology (no central server). Note that FedAvg used
in the proposed method is a centralized training with decentralized datasets. Please clarify the
difference in revision, making sure the readers understand which aspect is decentralized, the
datasets, the distributed optimization algorithm, or just computation.
***11. Reproducibility***
The proposed algorithm is complex. Without the code release, it’s hard to check the correctness
of the implementation. Deep learning normally costs time, I am curious to know how the authors
implement such a complex system/algorithm which can train so many nodes in parallel. If it is a
simulator, the training time will be too long since so there so many nodes (300+) with DNN
models. If the authors can provide code, maybe all my confusion can be addressed, and I am
willing to increase my rating.
***Suggestions***
(1) I encourage the authors demonstrate the idea with more realistic datasets. For example,
in social network, it is possible to run models on the smartphones of Internet users to
protect their privacy, which is much better than the transportation example.
(2) In the current version, there is no contribution in modeling. Thus, I suggest authors to
think about customizing the model to obtain benefit of efficiency and privacy.
(3) More intuition and discussion are requires for the training method. Why alternating
minimization works well in the GCN setting.
(4) Since the training results are not convincing, I suggest authors run more experiments to
understand the proposed method and explain why there are some counter-intuitive
results.
(5) Privacy is not the main focus, but some discussions and related works should be
mentioned.
Overall Comments
Due to so many concerns mentioned above, I encourage the authors to have a deeper
understanding of federated learning and address these issues in the revision, especially the
motivation, and demonstrating the effectiveness of the proposed alternating training method.
After thinking a while, I cannot find a practical scenario that requires GCN-based node-level
privacy. At least, the sensor dataset is not a good example. Maybe the authors can search for a
practical dataset to demonstrate motivation.
In addition, please focus on just one gist rather than intertwine too many aspects and claim all
contributions without any trade-off discussion. Besides the training framework, it’s also
interesting research if we can see newly developed model architectures can trade-off many
important aspects in FL.

---

> ### Author Response · Authors · 2020-11-21
> **Response to AnonReviewer3 (Part1)**
>
> Thank you for your comprehensive review and suggestions to improve the paper. Below are our responses to your comments:
>
> **Motivation** (In reply to the reviewer’s point 1)
>
> > **The aim of our paper is to enable Graph Neural Network(GNN)-based architecture, which has shown its success in modeling centralized spatio-temporal data, under the setting of FL.** The primary challenge in FL is to respect constraints on data sharing and manipulation. These constraints can occur in scenarios where data contains sensitive information, such as financial data owned by different institutions. Due to the sensitivity of data, datasets from such scenarios are proprietary and hardly offer public access. Therefore, we demonstrate the applicability of our proposed model on the traffic dataset, which is a good representative example of data with spatio-temporal correlations, and has been extensively studied in spatio-temporal forecasting works without FL constraints [1-6]. Our proposed model is general and applicable to spatio-temporal datasets with sensitive information.
>
> > **The constraint of computation resources on edge devices is also a challenge in FL, and we address the problem by restricting the computation cost of on-device models.** In Section 4.1 and Table 3, we have shown that our proposed model achieves the best prediction performance while maintaining the low computation cost on edge devices. The reviewer also raises an important issue regarding the deployment of deep learning (DL) models. Light-weight DL models have been deployed on edge devices with general-purpose CPUs [7], and advancement of AI at the Edge [8] also offers promising solutions for model deployment on low-cost edge devices.
>
> **Our contribution** (In reply to the reviewer’s point 2)
>
> > The focus of our work is to propose a model that captures complex spatio-temporal relationships under the FL constraint. The FL constraint brings additional challenges in optimization and computation cost to models for centralized data. These challenges are not trivial and existing solutions, such as Split Learning (SL), cannot address them. We address these challenges by combining alternating optimization and Federated Averaging (FedAvg), and results in Section 4.3 show its superiority in prediction performance and communication cost. To the best of our knowledge, we propose the first GNN-based model for spatio-temporal prediction in the FL context.

---

> > ### Author Response · Authors · 2020-11-21
> > **Response to AnonReviewer3 (Part 2)**
> >
> > **Experimental Results** (In reply to the reviewer’s point 3)
> >
> > > **We use different model architectures for training with centralized data and training with the FL constraint (including Split Learning), thus we expect that they have different performance**. For a network of N sensors, the model trained with centralized data only has 1 encoder-decoder module and 1 Graph Network (GN) module, while the model trained with FL constraint has N different encoder-decoder modules and 1 GN module. When training a model with centralized data, it’s a common practice [1-6] for GCN+RNN-based models to let all nodes share the same RNN encoder-decoder. But with the FL constraint, since the training of an encoder-decoder must have access to one node’s raw input and labels, each node has to train its own encoder-decoder without direct access to the data from other nodes. Therefore, *”Split learning performs differently from training with centralized data”* is expected in our setting. We have also re-examined the experiments and corrected one problem in the calculation of the validation loss of the Split Learning baseline in the rebuttal version. However, our analysis of the difference of model architectures for training with centralized data and split learning still applies.
> >
> > > **We have also added as another baseline a variant of split learning that synchronizes the weights of encoder-decoder modules periodically with FedAvg (**SL+FedAvg**) to Section 4.3 in the rebuttal revision**. However, simply combining FedAvg with split learning shows inconsistent results on different datasets and its performance is always inferior to our proposed AT + FedAvg.
> >
> > > Figure 2 shows the validation loss during training of different training strategies on PEMS-BAY and METR-LA datasets, and Table 5 shows their prediction performance and the communication cost in training. We notice that (1) **AT + FedAvg consistently outperforms other baselines on both datasets.**. SL suffers from suboptimal prediction performance and high communication costs on both datasets; SL + FedAvg does not have consistent results on both datasets and its performance is always inferior to AT + FedAvg. (2) **AT + FedAvg has the lowest communication cost on METR-LA and the 2nd lowest communication cost on PEMS-BAY**, on which the baseline with the lowest communication cost (SL + FedAvg) has a much higher prediction error (4.383 vs 3.822). Both illustrate that our proposed training strategy, SL + FedAvg, achieves the best prediction performance as well as low communication cost compared to other baseline strategies.
> >
> > > **Our experimental results are in line with the reviewer's (and general) intuition that centralized training does indeed yield better performance, as we show in Table 2 and Table 5**. On both PEMS-BAY and METR-LA datasets, the GRU + GN model trained with centralized data has the lowest prediction error on the test set.
> >
> > > We have repeated the experiments on the PEMS-BAY dataset. We add corresponding results and analysis to Section 4.3 in the rebuttal revision. Results on PEMS-BAY dataset match our observations on METR-LA data.
> >
> > **Communication and computational cost** (In reply to the reviewer’s point 4)
> >
> > > FLOPS means “floating point operations per second” and is a common metric of computation cost/performance. Typical edge devices with general-purpose CPUs have around 1 GFLOPS (such as Raspberry Pi 2: 1.47 GFLOPS) [9]. In Table 3 we show that our proposed CNFGNN only needs to run a light-weighted model on each device with computation cost of around 1/10 of the cost of a heavy model while getting better prediction performance, which means the computation of the light-weighted model is 10 times faster than the heavy one. The deployment of DL models on edge devices is not the focus of our paper.
> >
> > **Implementation and Reproducibility** (In reply to the reviewer’s points 5 and 11)
> >
> > > We have released the code to reproduce our experimental results as the supplementary material. We simulate the execution of all federated learning algorithms over 300+ nodes on a single machine. Due to the resource constraint on a single machine, instead of running the training on all nodes fully in parallel, we partition nodes into several workers, and each worker trains its assigned nodes sequentially. Our simulation can demonstrate the prediction performance of federated learning algorithms. However, the real-world training time will be different from that of the simulation. We have provided the communication cost of baselines and our method in Table 3 and Table 5. We have also provided the size of exchange information and other necessary parameters for computing the communication cost in Table A1-A6 in the appendix.

---

> > > ### Author Response · Authors · 2020-11-21
> > > **Response to AnonReviewer3 (Part 3)**
> > >
> > > **Related Works** (In reply to the reviewer’s points 6, 8, and 9)
> > >
> > > > **Scalability**. Using sampling strategy for large-scale graph learning is also an important and open problem in the graph learning domain [10-13]. Applying existing GNN models with sampling strategies and integrating them into our proposed framework is an interesting research direction to extend our work. We have added corresponding references to Section 2 and discussion to Section 5 in the rebuttal revision.
> > >
> > > > **Privacy**. Various existing privacy-preserving techniques, such as Differential Privacy (DP), Secure Multi-Party Computation (MPC) and Homomorphic Encryption (HE) can potentially be integrated into our proposed framework and help build federated learning systems following the privacy-in-depth principle. Extending our framework with existing privacy-preserving methods will be an interesting further step towards a fully practical federated learning system as discussed in Section 2 and Section 5.
> > >
> > > > **Alternating Optimization**. Alternating optimization is a popular choice in non-convex optimization [14-17]. In the context of Federated Learning, [18] uses alternating optimization for learning a simple global model and reduces the number of communicated parameters, and [19] uses alternating optimization for knowledge distillation from server models to edge models. In our work, we utilize alternating optimization to effectively train on-device modules and the server module jointly, which captures temporal and spatial relationships respectively. We thank the reviewer for pointing out related works in alternating optimization, and we have added corresponding references to Section 2.
> > >
> > > **Non-IID Property of the Dataset** (In reply to the reviewer’s point 7)
> > >
> > > > We show the histograms of traffic speed on different nodes of PEMS-BAY and METR-LA in Figure A2, Appendix A.4.  For each dataset, we only show the first 100 nodes ranked by their IDs for simplicity.  The histograms show that the data distribution varies with nodes, thus data on different nodes are not independent and identically distributed.
> > >
> > > > Assuming there are edge servers distributed in multiple regions, each of which can aggregate data collected on nearby sensors, is a practical setting. In this case, each edge server will have more data for training a better deep learning model, and our proposed CNFGNN is also applicable to run on edge servers for capturing the spatial relationships among them.
> > >
> > > **Clarification of some understanding in federated learning** (In reply to the reviewer’s point 10)
> > >
> > > > (1) **Improving latency**. A fully decentralized training framework will have advantage in terms of inference latency since edge devices can run the inference locally instead of waiting for the inference results from a central server. Thus we consider extending our current framework to a fully decentralized framework for our future work.
> > >
> > > > (2) **CNFGNN is for decentralized data.** We thank the reviewer for this astute observation and we have corrected the corresponding description of Federated Learning in the abstract. We clarify here that our method CNFGNN addresses the case of decentralized data. Extending it to a fully decentralized learning framework is a promising direction for our future work, as we have discussed in Section 5.
> > >
> > > **Our Reply to the Reviewer’s Suggestions**
> > >
> > > > (1) See the paragraph **Motivation**.
> > >
> > > > (2) See the paragraph **Our Contribution**.
> > >
> > > > (3)-(4) See the paragraph **Experimental Results**.
> > >
> > > > (5) See the paragraph **Related Works** and **Privacy**.
> > >
> > > Thanks again for your comprehensive review and suggestions.

---

> > > > ### Author Response · Authors · 2020-11-21
> > > > **Response to AnonReviewer3 (Part 4)**
> > > >
> > > > References:
> > > >
> > > > [1]. Yaguang Li, Rose Yu, Cyrus Shahabi, and Yan Liu. Diffusion convolutional recurrent neural network: Data-driven traffic forecasting. In International Conference on Learning Representations (ICLR ’18), 2018.
> > > >
> > > > [2]. Bing Yu, Haoteng Yin, and Zhanxing Zhu. Spatio-Temporal Graph Convolutional Networks: A Deep Learning Framework for Traffic Forecasting. In Proc. of IJCAI, 2018.
> > > >
> > > > [3]. Zonghan Wu, Shirui Pan, Guodong Long, Jing Jiang, and Chengqi Zhang. Graph WaveNet for Deep Spatial-Temporal Graph Modeling. In Proc. of IJCAI, 2019.
> > > >
> > > > [4]. Weiqi Chen, Ling Chen, Yu Xie, Wei Cao, Yusong Gao, and Xiaojie Feng. Multi-Range Attentive Bicomponent Graph Convolutional Network for Traffic Forecasting. In Proc. of AAAI, 2019.
> > > >
> > > > [5]. Chuanpan Zheng, Xiaoliang Fan, Cheng Wang, and Jianzhong Qi. GMAN: A Graph Multi-Attention Network for Traffic Prediction. In Proc. of AAAI, 2020.
> > > >
> > > > [6]. Wu, Zonghan, et al. Connecting the Dots: Multivariate Time Series Forecasting with Graph Neural Networks. In Proc. of KDD, 2020.
> > > >
> > > > [7]. https://blog.openmined.org/federated-learning-of-a-rnn-on-raspberry-pis/
> > > >
> > > > [8]. https://developer.nvidia.com/embedded-computing
> > > >
> > > > [9]. http://web.eece.maine.edu/~vweaver/group/green_machines.html
> > > >
> > > > [10]. Hamilton, Will, Zhitao Ying, and Jure Leskovec. "Inductive representation learning on large graphs." Advances in neural information processing systems. 2017.
> > > >
> > > > [11]. Ying, Rex, et al. "Graph convolutional neural networks for web-scale recommender systems." Proceedings of the 24th ACM SIGKDD International Conference on Knowledge Discovery & Data Mining. 2018.
> > > >
> > > > [12]. You, Jiaxuan, Rex Ying, and Jure Leskovec. "Position-aware Graph Neural Networks." ICML. 2019.
> > > >
> > > > [13]. Huang, Wenbing, et al. "Adaptive sampling towards fast graph representation learning." Advances in neural information processing systems. 2018.
> > > >
> > > > [14]. Alekh Agarwal, Animashree Anandkumar, Prateek Jain, Praneeth Netrapalli, and Rashish Tandon.Learning sparsely used overcomplete dictionaries.  InConference on Learning Theory, pp. 123–137, 2014.
> > > >
> > > > [15]. Sanjeev Arora, Rong Ge, and Ankur Moitra. New algorithms for learning incoherent and overcom-plete dictionaries. InConference on Learning Theory, pp. 779–806, 2014.
> > > >
> > > > [16]. Sanjeev Arora, Rong Ge, Tengyu Ma, and Ankur Moitra.  Simple, efficient, and neural algorithms for sparse coding. 2015.
> > > >
> > > > [17]. Prateek Jain and Purushottam Kar.   Non-convex optimization for machine learning. Foundations And Trends R© in Machine Learning, 10(3-4):142–363, 2017.   ISSN 1935-8237.   doi:  10.1561/2200000058. URLhttp://dx.doi.org/10.1561/2200000058.
> > > >
> > > > [18]. Paul Pu Liang, Terrance Liu, Liu Ziyin, Ruslan Salakhutdinov, and Louis-Philippe Morency. Thinklocally,  act globally:  Federated learning with local and global representations.arXiv preprint arXiv:2001.01523, 2020.
> > > >
> > > > [19]. Chaoyang He, Salman Avestimehr, and Murali Annavaram. Group knowledge transfer: Collabora-tive training of large cnns on the edge. NeurIPS 2020.

---

> ### Author Response · Authors · 2020-11-23
> **Thanks for your comprehensive review! Looking forward to a fruitful discussion.**
>
> We thank you for your comprehensive and helpful review and suggestions. We have addressed your concerns in our replies and have updated the submission draft according to your suggestions. We are looking forward to having an in-depth and fruitful discussion so that we can clarify any further confusion or concerns.

---

### Official Review · AnonReviewer1 · 2020-11-02
**The paper is okay to me**

**Rating:** 6
**Confidence:** 1

**Review:**

Graph neural networks and federated learning are both promising directions of works individually. This papers is apparently one of the first few attempts to combine them for spatio-temporal data modeling. The time series data in the local nodes is modelled by an Encoder-Decoder architecture and spatial locality property of various nodes is captured by the server. The Encoder at each node projects the time series data into an embedding space. This embedding is used by the GNN at the server as node features. The server side GNN outputs node embeddings. The Encoder embeddings and the GNN embeddings are then concatenated and fed to the decoder that predicts the outputs for the subsequent time steps. To ensure that all the nodes encode their temporal data in a common space, the encoders are shared by the clients. Overall, the results look promising.

Following are some doubts / concerns I have:

1. In Eq. 3, the specifics of how the edge features and global features are extracted is never mentioned in the paper. All that is mentioned is how they connect two nodes based on their distance of separation.

2. Can authors please elaborate more on Step 25 of Algorithm 1? Given \delta h_{G,c,i}, and l_i, can we compute \delta \theta_{G_N} Overall, how the parameters of GN are updated is not very clear to me.

3. In Algorithm 2, why is the client sending gradient of node embeddings back to the server?

4. In Table 3, why are we not comparing the computational cost on the server?

---

> ### Author Response · Authors · 2020-11-18
> **Response to AnonReviewer1**
>
> Thank you for your comments and suggestions to improve the paper. Below are our responses to your comments:
>
> 1. *“In Eq. 3, the specifics of how the edge features and global features are extracted is never mentioned in the paper.”*
>
> > For the first layer of Graph Network(GN), we set the edge feature on edge $(i,j)$ as the value at $(i,j)$ of the adjacency matrix, and global features as empty vectors (which means the first layer only utilizes node features and the adjacency matrix as input) since the dataset does not contain global features with explicit meanings. For each of the following layers of GN, we use the output from the previous layer as its input, which contains updated edge/node/global embeddings. We have added further clarification highlighted with red in Section 3.2.2 of the rebuttal revision.
>
> 2. *“Can authors please elaborate more on Step 25 of Algorithm 1?”*
>
> > We use step 25 to calculate the summation of gradients w.r.t parameters of the server-side model of losses over all nodes. Using the idea of back-propagation, $h_{G,c,i}$.backward($\nabla h_{G,c,i} \ell_i$) returns $\nabla_{\theta_{GN}}\ell_i$. Since $\ell = \sum_{i \in \mathcal{V}} \ell_i$,  $\nabla_{\theta_{GN}}\ell = \sum_{i \in \mathcal{V}} \nabla_{\theta_{GN}}\ell_i$. We have added further clarification in Step 23-27 of Algorithm 1 in the rebuttal revision.
>
> 3. *“In Algorithm 2, why is the client sending gradient of node embeddings back to the server?”*
>
> > In our federated learning architecture, the labels are only available on the client's side. As a result, to evaluate the gradient of the overall loss w.r.t. the parameters on the server-side using backpropagation, the clients need to communicate the gradients of the node embeddings back to the server.
>
> 4. *“In Table 3, why are we not comparing the computational cost on the server?”*
>
> > The primary challenge in Federated Learning is to minimize the computation cost on the edge devices, since these usually have constraints on energy utilization [1,2]. As a result, the computations on the client side are more relevant for this application. To address this challenge, our proposed model (CNFGNN) captures complex spatio-temporal relationships among multiple nodes on the server side model instead of increasing the complexity of client side models. As demonstrated in Table 3, the proposed model (CNFGNN) provides superior forecasting performance without incurring additional computation cost.
>
> References:
>
> [1]. Shi, W., & Dustdar, S. (2016). The promise of edge computing. Computer, 49(5), 78-81.
>
> [2]. Khanna, A., Sah, A., & Choudhury, T. (2020, April). Intelligent Mobile Edge Computing: A Deep Learning Based Approach. In International Conference on Advances in Computing and Data Sciences (pp. 107-116). Springer, Singapore.

---

### Author Response · Authors · 2020-11-18
**Code Release**

We have released the code as the supplementary material:
https://openreview.net/attachment?id=HWX5j6Bv_ih&name=supplementary_material

---

### Decision · Program_Chairs · 2021-01-07
**Final Decision**

**Decision:**

Reject

**Comment:**

This paper received mixed reviews: two positives (6, 6) and two negatives (5, 3). However, the positive reviewers have very low confidence, do not show strong supports for this paper. The reviewers raised various concerns about this paper, and there still exist remaining critical issues although the authors made substantial efforts to answer the questions.

After reading the paper and all the comments by the reviewers, I decided to recommend rejecting this paper mainly due to its weak technical contribution and ignorance of privacy issues. Note that this opinion is shared with two negative reviewers. The proposed model and alternative training scheme are straightforward, and the novelty is not distinct. Also, the authors seem to assume that "the extracted feature vectors and corresponding gradients are not sensitive". This comment is given by R2 but has not been clarified. The proposed method is lacking in this aspect and it is hard to say that it is an FL approach.